# Clinical and Functional Characterization of the Recurrent TUBA1A p.(Arg2His) Mutation

**DOI:** 10.3390/brainsci8080145

**Published:** 2018-08-07

**Authors:** Jennifer F. Gardner, Thomas D. Cushion, Georgios Niotakis, Heather E. Olson, P. Ellen Grant, Richard H. Scott, Neil Stoodley, Julie S. Cohen, Sakkubai Naidu, Tania Attie-Bitach, Maryse Bonnières, Lucile Boutaud, Férechté Encha-Razavi, Sheila M. Palmer-Smith, Hood Mugalaasi, Jonathan G. L. Mullins, Daniela T. Pilz, Andrew E. Fry

**Affiliations:** 1Institute of Medical Genetics, University Hospital of Wales, Cardiff CF14 4XW, UK; jennifer.gardner@wales.nhs.uk (J.F.G.); sheila.palmer-smith@wales.nhs.uk (S.M.P.-S.); hood.mugalaasi@wales.nhs.uk (H.M.); 2Division of Cancer and Genetics, School of Medicine, Cardiff University, Cardiff CF14 4XN, UK; cushiont@cardiff.ac.uk (T.D.C.); pilzdt@cardiff.ac.uk (D.T.P.); 3Paediatrics Department, Venizelion Hospital, Knossos Ave, P.O. Box 44, Heraklion, 714 09 Crete, Greece; niotakisg@yahoo.gr; 4Epilepsy Genetics Program, Department of Neurology, Division of Epilepsy and Clinical Neurophysiology, Boston Children’s Hospital, Boston, MA 02115, USA; heather.olson@childrens.harvard.edu; 5Fetal-Neonatal Neuroimaging and Developmental Science Center, Boston Children’s Hospital, Harvard Medical School, Boston, MA 02115, USA; ellen.grant@childrens.harvard.edu; 6Clinical Genetics Unit, Great Ormond Street Hospital for Children NHS Trust, Great Ormond Street, London WC1N 3JH, UK; richard.scott@genomicsengland.co.uk; 7Department of Neuroradiology, North Bristol NHS Trust, Frenchay Hospital, Bristol BS16 1LE, UK; neilstoodley@doctors.org.uk; 8Division of Neurogenetics, Hugo W. Moser Research Institute, Kennedy Krieger Institute, Baltimore, MD 21205, USA; cohenju@kennedykrieger.org (J.S.C.); naidu@kennedykrieger.org (S.N.); 9Department of Neurology, The Johns Hopkins Hospital, Baltimore, Maryland, MD 21287, USA; 10Department of Pediatrics, The Johns Hopkins Hospital, Baltimore, Maryland, MD 21287, USA; 11Unité d’Embryofœtopathologie, Service d’Histologie Embryologie Cytogénétique, Hôpital Necker-Enfants Malades, Assistance Publique Hôpitaux de Paris (APHP), 75004 Paris, France; tania.attie@inserm.fr (T.A.-B.); maryse.bonniere-darcy@aphp.fr (M.B.); lucile.boutaud@gmail.com (L.B.); ferechte.razavi@aphp.fr (F.E.-R.); 12Institut Imagine, INSERM U1163, Université Paris Descartes, Sorbonne Paris Cite, 75006 Paris, France; 13Institute of Life Science, Swansea University Medical School, Swansea SA2 8PP, UK; j.g.l.mullins@swansea.ac.uk; 14Department of Clinical Genetics, West of Scotland Regional Genetics Service, Queen Elizabeth University Hospital, Glasgow G51 4TF, UK

**Keywords:** *TUBA1A*, tubulin, p.(Arg2His), R2H, tubulinopathy, polymicrogyria, cerebellar hypoplasia

## Abstract

The *TUBA1A* gene encodes tubulin alpha-1A, a protein that is highly expressed in the fetal brain. Alpha- and beta-tubulin subunits form dimers, which then co-assemble into microtubule polymers: dynamic, scaffold-like structures that perform key functions during neurogenesis, neuronal migration, and cortical organisation. Mutations in *TUBA1A* have been reported to cause a range of brain malformations. We describe four unrelated patients with the same de novo missense mutation in *TUBA1A*, c.5G>A, p.(Arg2His), as found by next generation sequencing. Detailed comparison revealed similar brain phenotypes with mild variability. Shared features included developmental delay, microcephaly, hypoplasia of the cerebellar vermis, dysplasia or thinning of the corpus callosum, small pons, and dysmorphic basal ganglia. Two of the patients had bilateral perisylvian polymicrogyria. We examined the effects of the p.(Arg2His) mutation by computer-based protein structure modelling and heterologous expression in HEK-293 cells. The results suggest the mutation subtly impairs microtubule function, potentially by affecting inter-dimer interaction. Based on its sequence context, c.5G>A is likely to be a common recurrent mutation. We propose that the subtle functional effects of p.(Arg2His) may allow for other factors (such as genetic background or environmental conditions) to influence phenotypic outcome, thus explaining the mild variability in clinical manifestations.

## 1. Introduction

*TUBA1A* is a highly-conserved gene with few changes among eukaryotes and few polymorphic variants in human populations. *TUBA1A* encodes the tubulin alpha-1A chain, a protein that is highly-expressed in the cerebral cortex, hippocampus, cerebellum, and brainstem of the developing fetal brain, with a decrease in postnatal and adult stages [1,2]. Alpha- and beta-tubulin subunits form dimers that coassemble into microtubules. Microtubules are dynamic polymers that perform a range of mechanical tasks within the cell. As major components of the mitotic spindle, microtubules control division of neuronal progenitors to produce neurons. In turn, they generate the push-and-pull forces that are required for the migration of primitive neurons, from deep proliferative areas to the cortical plate. Subsequently, bundles of stable and polarised microtubule polymers generate long axons facilitating cortical organisation and synaptic connectivity.

*TUBA1A* was the first tubulin gene to be associated with human brain malformations [3]. Mutations in *TUBA1A* have been reported in patients with a range of brain malformations, including lissencephaly, microlissencephaly, polymicrogyria, and simplified gyri [4,5,6]. They have also been reported in patients with hydranencephaly-like dysplasias, cerebral palsy, and autistic spectrum disorders [7,8,9]. *TUBA1A* mutations (as with other tubulinopathies) are often associated with hypoplasia/agenesis of the corpus callosum, hypoplasia/dysplasia of the cerebellum, and dysmorphic basal ganglia [5,6]. Common clinical features in *TUBA1A* patients include microcephaly, intellectual disability, motor impairment, and epilepsy. Mutations in several other tubulin genes have been reported in patients with brain malformations including TUBB2B [10], *TUBB3* [11,12], *TUBB* [13], *TUBB2A* [14], and *TUBG1* [15]. However, *TUBA1A* mutations remain the most common cause of tubulin-related brain malformations, with over 60 mutations being described to date [6]. Most disease-causing *TUBA1A* mutations are de novo, although familial recurrence due to parental somatic mosaicism has been reported [4,16].

Pathogenic *TUBA1A* mutations have been found distributed throughout the gene. A handful of recurrent *TUBA1A* mutations have been reported. These include the p.(Arg402His) mutation, which has been reported in at least five patients and is associated with classic lissencephaly [3,17]. Similarly, the recurrent p.(Arg264Cys) mutation has been found in several patients and is typically associated with central pachygyria [3,18,19]. Few genotype-phenotype correlations have been reported for *TUBA1A*. However, the phenotypic effects of a specific recurrent mutation are generally consistent. Alpha-tubulin must fold in a precise way and present specific shapes and charges on its surface to interact with other proteins (e.g., beta-tubulin subunits, microtubule binding proteins) and to correctly handle and hydrolyse guanosine-5′-triphosphate (GTP). Many *TUBA1A* mutations have been shown to disrupt protein folding and/or heterodimer formation, resulting in either a reduced yield or reduced stability [20].

During the clinical diagnostic work-up of two unrelated patients with developmental delay and brain abnormalities, we identified the same mutation, c.5G>A, p.(Arg2His), in *TUBA1A*. To define the clinical consequences of this mutation, we collected detailed phenotype information from both patients and two additional patients that were previously reported in the literature [21,22,23]. We examined the functional impact of the mutation by in vitro microtubule studies and computer-based protein structure modeling.

## 2. Materials and Methods

### 2.1. Patients

Patients 1 and 2 were diagnosed during routine clinical diagnostic work-ups. Patient 1 underwent testing with a 12-gene polymicrogyria sequencing panel (targets enriched by Agilent SureSelect system, followed by Illumina sequencing) in the United States. Patient 2 had testing with a 40 gene cortical malformation gene panel (HaploPlex target enrichment system followed by Illumina sequencing) in the United Kingdom. The mutations in Patients 1 and 2 were confirmed and shown to be de novo by Sanger sequencing in the patient and both parents. Patient 3 underwent trio-based whole exome sequencing (WES) as a part of their routine clinical diagnostic work-up in the United States [21,23]. The approach to analysis and filtering of the WES data has previously been described [21]. No other candidate variants were identified in the patient. Patient 4 underwent targeted sequencing of a panel of 423 genes that are associated with corpus callosum anomalies in France [22]. The approach to analysis and filtering of this panel has previously been described [24,25]. No other candidate variants were identified in the patient. Consent was obtained from the parents of all the participants for publication. The genomic location of the mutation is chr12:g.49580615C>T (GRCh37/hg19), rs587784491. Coding and protein positions of *TUBA1A* mutations are based on GenBank accession codes NM_006009.3 (ENST00000301071.7) and NP_006000.2, respectively.

### 2.2. Homology Modelling

Structural predictions of wild-type and mutant TUBA1A protein subunits were generated while using a previously-described homology modeling pipeline [26]. This approach uses the solved structure of a homologous template to predict the folding of a target sequence. The target sequence was wild-type TUBA1A (NP_006000.2). The template used was the crystal structure of Tubulin alpha-1B from *Bos taurus* (Protein Data Bank (PDB): 4I4T) [27], which shares 99% sequencing identity with human TUBA1A. Microtubule architecture was based on a previously published template (PDB: 2XRP) [28]. Homology modelling was performed by MODELLER (version 9.17) [29]. Structural models were viewed and analysed while using the UCSF Chimera software (version 1.12) [30,31].

### 2.3. Expression Construct Mutagenesis and Cell Culture

A C-terminally FLAG-tagged wild-type TUBA1A expression construct (pRK5-TUBA1A-C-FLAG) was modified to generate TUBA1A-R2H by site-directed mutagenesis using the QuikChange mutagenesis kit (Stratagene, La Jolla, CA, USA). HEK-293 cells were cultured in Dulbecco’s modified Eagle’s media (ThermoFisher, Waltham, MA, USA, catalogue number 41966029), supplemented with 10% fetal calf serum (ThermoFisher, 10500056) and 1% penicillin/streptomycin (ThermoFisher, 15070063), as previously described [14].

### 2.4. Immunocytochemistry

HEK-293 cells were cultured in Dulbecco’s modified Eagle’s media (ThermoFisher, 41966029). supplemented with 10% fetal calf serum (ThermoFisher, 10500056) and 1% penicillin/streptomycin (ThermoFisher, 15070063) and incubated at 37 °C 5% CO_2_. Cells were seeded onto poly-D-Lysine (Sigma-Aldrich, St. Louis, MO, USA, P6407) pre-coated 13 mm glass coverslips. After 24 h, the cells were transfected with either wild-type or mutant expression constructs using Lipofectamine 2000 (ThermoFisher, 11668030). Twenty-four hours post-transfection, the cells were fixed with methanol at −20 °C for five minutes. Fixed cells were blocked with blocking buffer (phosphate-buffered saline (PBS) with 2% Bovine Serum Albumin (BSA; Sigma-Aldrich, B4287) and 0.5% Triton (Sigma-Aldrich, T8787)) for 30 min at room temperature (23 °C). Cells were immunostained with rabbit anti-FLAG (Sigma-Aldrich, F7425; 1:500) and mouse anti-alpha-tubulin (Sigma-Aldrich, T6199; 1:750) diluted in PBS with 2% BSA and 0.1% Triton for one hour at room temperature. Primary antibodies were aspirated, cells washed three times with PBS, and incubated with AlexaFluor^568^-conjugated goat anti-rabbit (ThermoFisher, A11011) and AlexaFluor^488^-conjugated goat anti-mouse (ThermoFisher, A11011) secondary antibodies for 30 minutes at room temperature, and protected from light from this point onwards. Cells were rinsed with PBS, mounted onto glass slides with ProLong Gold mounting medium (ThermoFisher, P10144) and stored at 4 °C until examined by confocal microscopy (Zeiss Axioscope).

### 2.5. Predicting the Probability of TUBA1A Substitutions

The genomic DNA sequence of the *TUBA1A* gene (based on transcript ENST00000301071.7) was obtained from the Ensembl Genome Browser [32]. A sliding window was implemented using a Perl script. For each 7-nucleotide window the script recorded the position and base of the central nucleotide. The heptanucleotide sequence was then looked up in the data from [33] (Supplementary Table 7 from that paper). The substitution probabilities for changing the central nucleotide to each of the three alternative bases were taken (averaging African, Asian, and European values). The cDNA and protein consequences of each substitution were derived using Mutalyzer [34,35]. Predicted substitution probabilities were obtained for all coding positions, introns (±20 base pairs flanking exons), and 5’ and 3’ untranslated regions (±20 base pairs).

## 3. Results

### 3.1. Clinical Features of Patients with the p.(Arg2His) Mutation

We identified two unrelated patients (Patients 1 and 2) with the same *TUBA1A* missense mutation, c.5G>A, p.(Arg2His). A search of the literature found reports of two additional patients with the p.(Arg2His) mutation (Patients 3 and 4) [21,22,23]. Only brief descriptions of the two published subjects were previously available. We obtained detailed clinical information from the four individuals (Table 1, detailed case reports are provided in the Appendix A). All four mutations were de novo. Consistent features in the living patients were developmental delay and microcephaly. MRI brain images from Patients 1–3 were available for review (Figure 1). The images demonstrated the hypoplasia and dysplasia of the cerebellar vermis (3/3), hypoplasia or dysgenesis of the corpus callosum (3/3), and dysmorphic basal ganglia (3/3). Patient 1 had bilateral perisylvian polymicrogyria. The pons of all three patients was small, particularly affecting the belly of the pons.

Patient 4 was a fetus terminated at 36 weeks gestation. Post-mortem examination of Patient 4 found a small brain (weight on 5th centile) with shortening of the corpus callosum and cerebellar hypoplasia (Figure 2A,B). Neuropathology examination found bilateral perisylvian polymicrogyria (Figure 2C–E). At the supratentorial level, callosal fibers and corticospinal tracts (CST) were hypoplastic. The brainstem was shortened and dysmorphic, displaying a Z-shaped kink. At the level of the cerebral peduncles, the CST were present but reduced in size. The pons was reduced in size in its basilar part. In the pons the CST were present at the junction with the peduncles but showed a chaotic pattern in between the pontine nuclei. The transverse pontine fibers were also reduced, and associated with cerebellar heterotopias and hypoplastic deep nuclei. At the level of the medulla, the pyramids were present but hypoplastic. The inferior olivary nuclei were also reduced in size. Neuronal heterotopia of the olivary nuclei was noted. At the cervical spinal cord level, crossing CST were absent. Cerebellar foliation was normal, but lamination was impaired with rare and misaligned Purkinje cells.

### 3.2. Modelling the Structural Effects of p.Arg2His

The Arg2 residue of TUBA1A is highly conserved across species and tubulin isoforms (Appendix A). The p.(Arg2His) variant is not present in gnomAD and multiple in silico prediction tools suggest it is deleterious (Appendix A). However, the physicochemical difference between arginine and histidine is relatively small (Grantham difference 29) with both of the residues having positively-charged side chains. The c.5G>A change is predicted to have minimal effects on the splicing at the adjacent splice acceptor site (Appendix A). When incorporated into polymerised microtubule, the N-terminus of alpha-tubulin is positioned near the inter-dimer interface, between the alpha-tubulin subunit of one heterodimer and the beta subunit of the next heterodimer. To study the effects of p.Arg2His on the three-dimensional structure of the protein, we compared wild-type and mutant TUBA1A by modelling the alpha/beta-tubulin heterodimer (Figure 3A,B) (the protein variant is given here without brackets as we know the amino acid sequence in a simulation). The effects of the mutation were mild. No predicted hydrogen bonding was lost or gained between the alpha- and beta-tubulin subunits as a result of p.Arg2His. A hydrogen bond between Arg2 and the highly-conserved Cys4 residue within TUBA1A was lost. In addition, new bonds between Glu3, and both Asn50 and Thr130 were predicted to form as a result of the substitution. Additional conformational changes were predicted to occur in a loop region (Asp38 to Asn51, Figure 3B), which may affect interactions between heterodimers.

### 3.3. Heterologous Expression of TUBA1A-R2H in HEK-293 cells

TUBA1A containing the p.(Arg2His) mutation (TUBA1A-R2H) was expressed in cultured HEK-293 cells. TUBA1A-R2H incorporated into the microtubule polymer network (Figure 3C), suggesting that it successfully folds and dimerises with endogenous beta-tubulin. However, in comparison to wild-type TUBA1A (Figure 3D), there was a slight increase in the proportion of the mutant FLAG-tagged protein seen unpolymerised within the cytosol. This suggests the mutation subtly alters the function (folding, dimerisation, or coassembly) of the subunit, but that once incorporated the dynamics of the mutant subunit are similar to wild type.

### 3.4. Substitution Probability of Recurrent TUBA1A Mutations

The observation of p.(Arg2His) on four separate occasions suggested that it was a common recurrent *TUBA1A* mutation. However, we noted that p.(Arg2His) had not been reported in previous large tubulinopathy cohorts [6]. In contrast, several *TUBA1A* mutations have been found recurrently in tubulinopathy patients. Examples include p.(Arg214His) [6,36], p.(Arg264Cys) [3,18,19], p.(Arg390Cys) [17,37], p.(Arg402His) [3,17,38,39], p.(Arg402Cys) [17,19], and p.(Arg422His) [17,18,40]. This made us wonder whether p.(Arg2His) had a lower mutation rate than the other recurrent *TUBA1A* mutations or whether it was just ascertained less frequently. We observed that the recurrent mutations all occurred at CpG sites, which are prone to spontaneous deamination (CGx is the codon for arginine). This highlighted that sequence context was likely to be an important factor. To predict the substitution rates at these sites and to compare them with the rest of *TUBA1A*, we estimated the probability of all possible single-base substitutions in *TUBA1A* based on heptanucleotide context (target position and three flanking nucleotides either side). Heptanucleotide context has been shown to explain >81% of variability in substitution probabilities [33]. We found the seven recurrent *TUBA1A* mutations all ranked in the top 1% for substitution probability. The p.(Arg2His) mutation was the second highest in the group (ranking 7th out of 4548 possible substitutions) (Appendix A). These results suggest p.(Arg2His) has a mutation rate that is similar to other recurrent *TUBA1A* mutations. The lack of observations in previous tubulinopathy cohorts may therefore reflect differences in ascertainment.

## 4. Discussion

In this report, we describe four patients with the TUBA1A p.(Arg2His) mutation. The patients had similar phenotypes with mild variability. Shared features included developmental delay, microcephaly, hypoplasia, and dysplasia of the cerebellar vermis, dysplasia or thinning of the corpus callosum, and dysmorphic basal ganglia. The pons tended to be small, disproportionally affecting the belly of the pons. We suspect the pons is dyspastic (i.e., abnormally developed) as well as small. Histopathological abnormalities of the pons were noted in patient 4. Two of the patients had bilateral perisylvian polymicrogyria. These features are typical of a tubulinopathy spectrum disorder [6]. Our findings suggest that p.(Arg2His) is a common recurrent *TUBA1A* mutation. Tubulinopathy patients are often ascertained due to cortical malformations (e.g., the classical lissencephaly associated with the recurrent p.(Arg402His) mutation)). In contrast, p.(Arg2His) does not cause an extensive cortical malformation. This may explain why p.(Arg2His) has not been observed in previous tubulinopathy cohorts [6].

Phenotypic variability that is associated with recurrent *TUBA1A* mutations has previously been noted. For example, the p.(Arg390Cys) mutation was first reported in a patient with mild gyral simplification, complete agenesis of the corpus callosum, and cerebellar hypoplasia [17]. It was subsequently reported in a patient with asymmetrical perisylvian polymicrogyria, hypoplasia of the corpus callosum, dysplastic cerebellar vermis, dysmorphic basal ganglia, and severe hypoplasia of brainstem [37]. Similarly, p.(Arg214His) was initially reported in a fetus with central polymicrogyria-like cortical dysplasia, complete agenesis of the corpus callosum, and normal cerebellum [6]. It was then reported in a patient with diffuse irregular gyration and sulcation of the cortex, partial agenesis of the corpus callosum, hypoplasia of the cerebellar vermis, and globular thalami [36]. As with p.(Arg2His), these descriptions suggest variability, but with overlap in key elements of the phenotype (abnormalities of the cortex, corpus callosum, cerebellum, and basal ganglia). Some of the variability may be due to differences in the interpretation of the brain imaging. However, differences in genetic background, environmental factors, or random chance may also contribute. Oegema et al. [36] found that p.(Arg214His) caused only a mild functional deficit (incorporating into microtubule polymers at comparable levels to wild type but at a reduced rate) and subtle predicted structural effects. Mutations with relatively mild functional effects (such as p.(Arg214His) or p.(Arg2His)) may allow for other factors to influence phenotype outcome.

Mutations of the homologous Arg2 residue in other tubulin isoforms have been linked to human disease phenotypes. TUBB8 is the main beta-tubulin of oocytes. The p.(Arg2Lys) mutation in TUBB8 has been found to cause arrest of oocyte meosis [41]. The mutation is thought to affect folding of the protein as well as the assembly and stability of heterodimers. The p.(Arg2Met) mutation in TUBB8 has also been shown to cause arrest of oocyte maturation [42,43]. TUBB4A is a brain-expressed beta-tubulin isoform. A p.(Arg2Gly) mutation in TUBB4A has been identified in a family with dystonia type 4 (‘Whispering dysphonia’) [44,45]. TUBB4A p.(Arg2Trp) and p.(Arg2Gln) have been reported to cause hypomyelination with atrophy of the basal ganglia and cerebellum [46,47]. The Arg2 of TUBB4A is part of the MREI (Met-Arg-Glu-Ile) ‘auto-regulatory’ domain, which is involved in controlling the amount of the beta-tubulin produced by the cell. In addition, these mutations disrupt a salt bridge Arg2 forms with Asp249 in TUBB4A [48]. This salt bridge is not predicted to occur in TUBA1A as the homologous residues are further apart.

## 5. Conclusions

We have shown that the *TUBA1A* c.5G>A, p.(Arg2His) mutation causes cortical, callosal, and cerebellar abnormalities that are typical of tubulinopathy-associated brain malformations. Based on its sequence context (and observation in four unrelated patients), c.5G>A is likely to be a common recurrent mutation in *TUBA1A*. Our functional and computer modelling results suggest that p.(Arg2His) has subtle effects on microtubule function, possibly acting at the inter-dimer interface. We propose that the subtle functional effects of the mutation may allow for other factors (e.g., genetic background, environmental conditions, or random chance) to modulate outcome, explaining the mild phenotypic variability observed.

## Figures and Tables

**Figure 1 brainsci-08-00145-f001:**
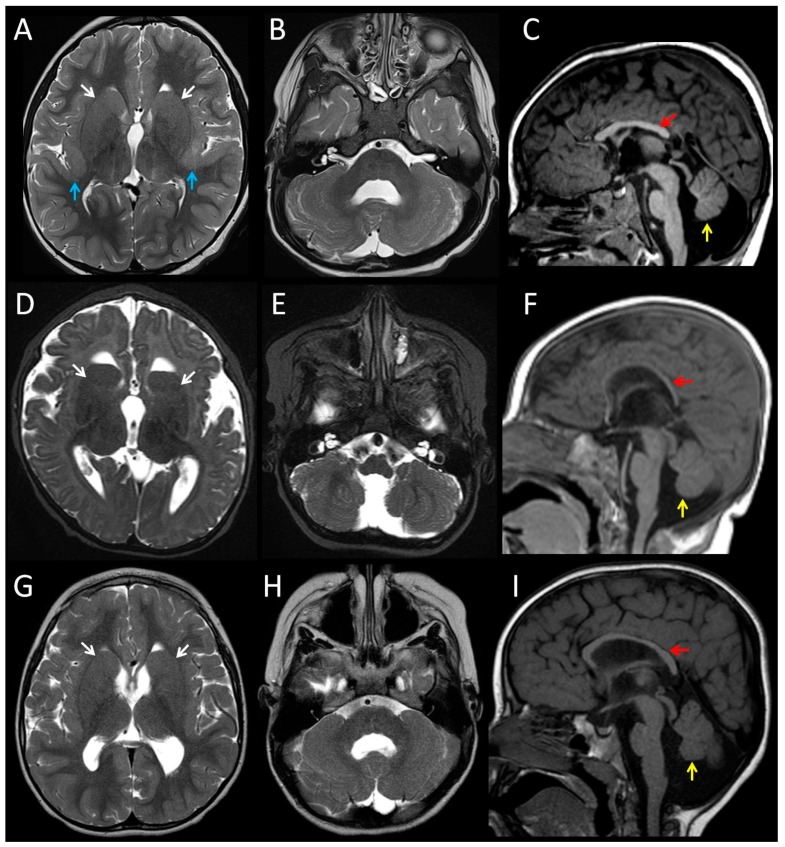
Magnetic resonance images from patients with the recurrent p.(Arg2His) TUBA1A mutation. T2-weighted axial and T1-weighted midline sagittal brain images for Patient 1 at age three years (**A**–**C**), Patient 2 at age six months (**D**–**F**), and Patient 3 at age 19 months (**G**–**I**). The images demonstrate hypoplasia and dysplasia of the cerebellar vermis (yellow arrows), thinning or partial agenesis of the corpus callosum (red arrows), globular basal ganglia with incomplete formation of the anterior limb internal capsule (white arrows), and bilateral perisylvian polymicrogyria (blue arrows). The pons is similar in size to the midbrain which suggests the pons is relatively small (**C**,**F**,**I**).

**Figure 2 brainsci-08-00145-f002:**
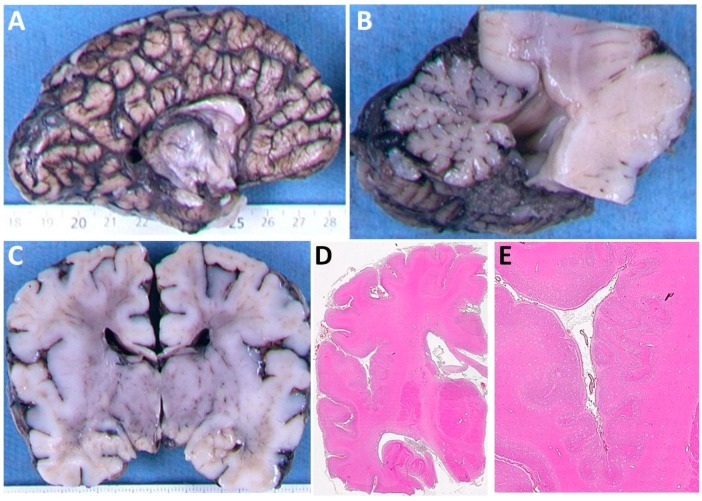
Neuropathology from Patient 4. (**A**) The medial aspect of right cerebral hemisphere showing a thin corpus callosom with absent rostrum. (**B**) Midline sagittal section of brain stem and cerebellum showing mild hypoplasia of the cerebellar vermis. (**C**) Coronal section of the cerebral hemispheres. The corpus callosum is thinned and there is thickening of the cortex around the sylvian fissures. (**D**) Stained section of the right cerebral hemisphere revealing abnormal folding of the cortical ribbon around the sylvian fissure. (**E**) A magnified view of (**D**) demonstrating polymicrogyria around the sylvian fissure.

**Figure 3 brainsci-08-00145-f003:**
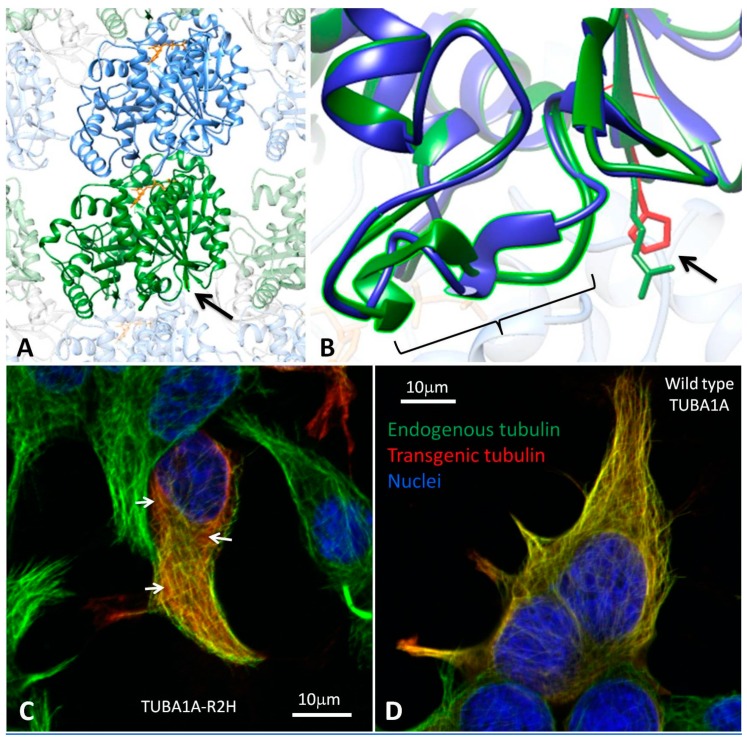
In silico modelling and in vitro functional analysis of the p.(Arg2His) mutation. (**A**) Ribbon models of alpha-tubulin (green) and beta-tubulin (blue) subunits aligned in a microtubule polymer. The position of Arg2 is shown (arrow) close to the inter-dimer interface (between alpha-tubulin and the beta-tubulin of an adjacent heterodimer). The mutation is on the opposite side of TUBA1A from the binding site of guanosine-5′-triphosphate (GTP, orange). (**B**) A close-up view of the Arg2 residue (arrow) with the mutant (purple ribbon, red side chain) and wild type (green ribbon and side chain) proteins superimposed. Only mild confirmation changes are predicted around the Arg2 residue. However, additional conformational changes are predicted between residues 38 and 51 (bracket). These may affect the interaction between heterodimers. (**C**) HEK-293 cells expressing FLAG-tagged TUBA1A-R2H. The cells are stained with DAPI (4′,6-diamidino-2-phenylindole, blue), anti-FLAG- (red), and anti-alpha-tubulin (green) antibodies. The microtubules appear yellow due to the colocalisation of endogenous (green) and FLAG-tagged transgenic (red) tubulin. The arrows indicate diffuse patches of transgenic mutant tubulin (red) in the cytoplasm between the microtubules. (**D**) Control cells expressing FLAG-tagged wild-type TUBA1A have less staining for the transgenic tubulin in the cytoplasm between the microtubules.

**Table 1 brainsci-08-00145-t001:** Clinical features of patients with the recurrent p.(Arg2His) TUBA1A mutation.

Patient	1	2	3	4 (fetus)
**Sex**	Male	Male	Male	Male
**OFC at Birth**	30 cm (−3.6 SD)	34 cm (−0.9 SD)	33 cm (−1.7 SD)	n/a
**Age at last review**	4 years	32 months	37 months	TOP at 36 weeks gestation
**Last OFC**	45 cm (−4.9 SD)	43 cm (−5.9 SD)	45 cm (−4.6 SD)	n/a
**Developmental delay**	Moderate	Severe	Moderate	n/a
**Seizures**	Yes (onset at 3 years)	Yes (onset at 12 months)	No	n/a
**Cerebral cortex**	Bilateral perisylvian polymicrogyria	Normal	Normal	Bilateral perisylvian polymicrogyria
**White matter**	Reduced	Reduced	Reduced	n/k
**Corpus callosum**	Partial agenesis	Thin	Thin	Short, no rostrum
**Basal ganglia**	Dysmorphic, prominent	Dysmorphic, prominent	Dysmorphic, prominent	n/k
**Cerebellum**	Hypoplasia and dysplasia of vermis	Hypoplasia and dysplasia of vermis	Hypoplasia and dysplasia of vermis	Hypoplasia, Impaired lamination, rare and misaligned Purkinje
**Brainstem**	Small pons	Small pons	Small pons	Neuronal heterotopia of olivary nuclei and hypoplastic pyramids

Key: *n/a/k* = not applicable/known; OFC = occipital frontal circumference; PMG = polymicrogyria; SD = standard deviations; TOP = termination of pregnancy.

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
