# Peer review of "Clinical and Functional Characterization of the Recurrent TUBA1A p.(Arg2His) Mutation"

_brainsci, 2018, doi:10.3390/brainsci8080145_

Round 1
Reviewer 1 Report
In this manuscript, Gardner and colleagues expand the range of TUBA1A variants found in human patients to associate with congenital brain malformations. The text is largely well written and data support most of the conclusions drawn. The rationale and addition to the story for the substitution work (section 3.4) is not clear at this point.
Specific comments:
references 1,2 seem to be supporting conclusions about TUBA1A expression. Ref 1 does not have any expression data and ref 2 only addresses expression in the dentate gyrus. This is a difficult issue to discuss in general as high degrees of similarity among tubulin proteins render antibodies nonspecific and make sequenced based analysis suspect. there is some limited data available from the tuba1a-lacZ transgenic construct mouse but not in the papers cited to support the summary statement made.
Ref 3 is meant to support autosomal dominance of a TUBA1A variant, but as the parent is a somatic mosaic, this should be more fully discussed/disclosed.
Patients 3 and 4 were analyzed by exome sequencing. Were there other candidate variants from this analysis? I don't doubt the veracity of the TUBA1A conclusion, but readers may want to know what else was in the exome or how it was filtered to the TUBA1A variants.
Fig3A - can you make the black arrow stand out more?
Fig3C,D - can you clearly label the panels so we can easily see which TUBA1A is expressed in each? The conclusion that TUBA1A-R2H creates more monomers free in the cytoplasm is not at all obvious from the images shown.
Section 3.4 - this is not all that well integrated into the rest of the paper. what is this analysis meant to tell us, how does it add to/complement the key message of the manuscript?
Author Response
Reply to Reviewer 2
Thank you for your comments.
Point 1: References 1,2 seem to be supporting conclusions about TUBA1A expression. Ref 1 does not have any expression data and ref 2 only addresses expression in the dentate gyrus. This is a difficult issue to discuss in general as high degrees of similarity among tubulin proteins render antibodies nonspecific and make sequenced based analysis suspect. there is some limited data available from the tuba1a-lacZ transgenic construct mouse but not in the papers cited to support the summary statement made..
The wrong references were cited. These have been changed to PMIDs 8891946 and 7996178 which provide more tuba1a-lacZ transgenic construct data and compare it to the expression of alpha-tubulin mRNA.
Point 2: Ref 3 is meant to support autosomal dominance of a TUBA1A variant, but as the parent is a somatic mosaic, this should be more fully discussed/disclosed.
The somatic mosaicism of the parent in that case is now highlighted in the introduction.
Point 3: Patients 3 and 4 were analyzed by exome sequencing. Were there other candidate variants from this analysis? I don't doubt the veracity of the TUBA1A conclusion, but readers may want to know what else was in the exome or how it was filtered to the TUBA1A variants.
We have queried this with the patients' clinicians. They have clarified that no other candidate variants were identified by the testing. This information has been added to the methods. Patient 4's testing was actually a large panel of genes associated with corpus callosum abnormalities - not whole exome sequencing. This has been corrected in the methods. Detailed descriptions of the analysis and filtering strategies for both patients have previously been published. The reference for Patient 3 was already included [PMID 25356970], the references for Patient 4 have been added [PMIDs 29681083, 29193896].
Point 4: Fig3A - can you make the black arrow stand out more?
Done
Point 5: Fig3C,D - can you clearly label the panels so we can easily see which TUBA1A is expressed in each? The conclusion that TUBA1A-R2H creates more monomers free in the cytoplasm is not at all obvious from the images shown.
We have:
(i) Labeled panels Fig3C and D with "TUBA1A-R2H" and "Wild type TUBA1A" respectively.
(ii) Added coloured text to indicate what the three stains target.
(iii) Added arrows to panel Fig3C indicating the patches of red (i.e. transgenic mutant tubulin) in the cytoplasm between the microtubules (which appear yellow due to colour addition).
(iv) Expanded the explanation in the Figure 3 legend to help the reader understand what is shown.
Point 6: Section 3.4 - this is not all that well integrated into the rest of the paper. what is this analysis meant to tell us, how does it add to/complement the key message of the manuscript?
We have re-written section 3.4 to better explain how it is integrated into the paper. The starting point for this section is the observation that - if R2H is a common recurrent mutation - why has it not turned up in previous big tubulinopathy patient studies? We propose two hypothesis, either (i) R2H is occurring less often than other recurrent mutations or (ii) R2H is being ascertained less often in traditional tubulinopathy patient studies. The similarity in predicted mutation rates with other recurrent mutations suggests that an ascertainment difference (probably due to the milder cortical phenotype) is more likely to be correct. We think the list of predicted rates for all possible TUBA1A substitutions will be a useful resources for future researchers studying recurrent TUBA1A mutations.
Reviewer 2 Report
This is a well written paper reporting the same de novo missense mutation in TUBA1A c.5G>A, p.(Arg2His) gene found by next generation sequencing in four unrelated patients with similar phenotype
The manuscript is well organized and documented; discussion is sound.
Furthermore same minor observations can be done:
Results/Supplementary Material:Case 1: What do you mean with “dysplasia of the pons”? Please describe.
Case 2: The sentence “The pons and was small but not hypoplastic " needs to be corrected or rewritten. Furthermore, the postulate pons defect (small but not hypoplastic) must be better described than the MRI (figure 1 D-F) is not evident.
Table 1:
Add fetus near Patient 4
Author Response
Reply to Reviewer 1
Thank you for your comments.
Point 1: Case 1: What do you mean with “dysplasia of the pons”? Please describe.
We suspect the pons is abnormally developed not just normally formed and small which hypoplastic implies. However, it is difficult to demonstrate dysplasia from the radiology alone - therefore we have:
i) Changed the description of Patient 1's brainstem in Table 1 [page 5] (to "Small pons")
ii) Changed the wording in the Supplementary Material [page 6] ("and a small pons").
iii) Added a section to the discussion explaining our thinking [page 9] "The pons tended to be small, disproportionally affecting the belly of the pons. We suspect the pons may be dyspastic (i.e. abnormally developed) as well as small. No specific histopathological abnormalities of the pons were noted in patient 4 but other brainstem structures (olivary nuclei and pyramids) were abnormal.")
Point 2: Case 2: The sentence “The pons and was small but not hypoplastic " needs to be corrected or rewritten. Furthermore, the postulate pons defect (small but not hypoplastic) must be better described than the MRI (figure 1 D-F) is not evident.
i) The wording has been corrected and simplified in the Supplementary Material [page 7] to "The pons was small."
ii) We have added text to the Figure 1 legend [page 6] "The pons is similar in size to the midbrain which suggests the pons is relatively small (C,F,I)."
Point 3: Table 1: Add fetus near Patient 4
Done